# Association between in-stent neointimal characteristics and native coronary artery disease progression

Hae Won Jung[1]◦, Chewan Lim[2]◦, Han Joon Bae[1], Jung-Hee Lee[3], Yong-Joon Lee[2], Jung-Sun Kim[2,4]*, Seung-Jun Lee[2,4], Sung-Jin Hong[2,4], Chul-Min Ahn[2,4], Byeong-Keuk Kim[2,4], Young-Guk Ko[2,4], Donghoon Choi[2,4], Myeong-Ki Hong[2,4,5], Yangsoo Jang[2,4,5]

1 Department of Cardiology, Daegu Catholic University Medical Center, Daegu, Korea, 2 Severance Cardiovascular Hospital, Yonsei University College of Medicine, Seoul, Korea, 3 Division of Cardiology, Yeungnam University Medical Center, Yeungnam University College of Medicine, Daegu, Korea, 4 Cardiovascular Institute, Yonsei University College of Medicine, Seoul, Korea, 5 Severance Biomedical Science Institute, Yonsei University College of Medicine, Seoul, Korea

◦ These authors contributed equally to this work.
* kjs1218@yuhs.ac

**Data Availability Statement:** All relevant data are within the manuscript and its Supporting Information files.

## Abstract

### Background and aims

The prognosis of stented lesions differs according to in-stent neointimal characteristics on optical coherence tomography (OCT). In particular, patients who show in-stent heterogeneous neointima are associated with a higher incidence of target lesion revascularization (TLR) compared with those who show in-stent non-heterogeneous neointima. However, the relationship between in-stent neointimal characteristics and native coronary atherosclerosis progression has not been clearly elucidated. The study aimed to investigate the relationship between in-stent neointimal characteristics and progression of native atherosclerosis.

### Methods

The neointimal characteristics of 377 patients with 377 drug-eluting stents (DESs) were quantitatively and qualitatively assessed using OCT. The OCT-based neointima was categorized as homogeneous (n = 207), heterogeneous (n = 93), and layered (n = 77). The relationship of non-target lesion revascularization (non-TLR) with neointimal characteristics was evaluated after OCT examination of the stents.

### Results

After a median follow-up duration of 40.0 months, patients with heterogeneous neointima showed significantly higher non-TLR rates than those with homogeneous neointima and tended to have higher non-TLR rates than those with layered neointima (heterogeneous vs. homogeneous:14.0% vs. 8.7%, p = 0.046; heterogeneous vs. layered neointima:14.0% vs. 7.8%, p = 0.152). Multivariate analysis showed that the independent determinants for non-TLR were heterogeneous neointima (HR: 2.237, 95% CI: 1.023–4.890, p = 0.044) and chronic kidney disease (hazard ratio [HR]: 8.730, 95% CI: 2.175–35.036, p = 0.002).

**Funding:** This research was supported by a grant from the Korea Health Technology R&D Project through the Korea Health Industry Development Institute (KHIDI), funded by the Ministry of Health & Welfare, Republic of Korea to JSK (No: HI15C1277), a grant from the National Research Foundation of Korea (NRF), funded by the Korean government (MSIT) to JSK (No. 2017R1A2B2003191), the Ministry of Science & ICT to SJK (2017M3A9E9073585), and the Cardiovascular Research Center (Seoul, Korea).

**Competing interests:** The authors have declared that no competing interests exist.

## Conclusions

The heterogeneous neointima in DES-treated lesions was associated with a higher incidence of non-TLR and target lesion failure. This finding suggests that the neointimal pattern may reflect the progression of the native lesion.

## Introduction

The prognosis of stented lesions may differ according to the in-stent neointimal characteristics observed on optical coherence tomography (OCT) [1–3]. In particular, patients who show in-stent heterogeneous neointimal growth are associated with a higher incidence of target lesion revascularization (TLR) compared with those who show in-stent non-heterogeneous neointimal growth [1, 2]. However, the relationship between in-stent neointimal characteristics and native coronary atherosclerosis progression has not been clearly elucidated. Therefore, in this study, we aimed to this study to investigate the relationship between in-stent neointimal characteristics and native atherosclerosis progression in untreated coronary segments using OCT in patients who underwent percutaneous coronary intervention (PCI) with drug-eluting stent (DES).

## Materials and methods

### Patient data selection

Among 540 patients with 576 stented lesions from the Yonsei OCT registry (Clinical Trials. gov NCT02099162), 163 patients were excluded from the study for the following reasons: 1) inadequate OCT image quality (n = 5), 2) index PCI with bare metal stents (n = 45), 3) OCT for two or more stented lesions (n = 36), 4) OCT after balloon angioplasty due to tight lesion (n = 5), 5) OCT-based evidence of neoatherosclerotic lesions (n = 39), 6) in-stent restenosis (ISR) lesions treated with a DES (n = 30) and 7) loss to follow-up (n = 3). Finally, 377 patients with 377 DES-treated lesions were included. The stent type was selected at the discretion of the physician at the time of index PCI. Among the patients included, 127 first-generation DESs and 250 second-generation DESs were used. A first-generation DES was defined as a sirolimus- (Cypher) and paclitaxel- (Taxus) eluting stent, and the second-generation DESs were defined as zotarolimus-(Endeavor Sprint or Resolute), everolimus- (Xience), and biolimus-(Nobori) eluting stents. DES implantation was performed using conventional techniques. Unfractionated heparin at 100 IU/kg was administered as an initial bolus, with additional boluses administered during the procedure to achieve an activated clotting time of 250–300 s. Dual antiplatelet therapy (100 mg aspirin and 75 mg clopidogrel, 180 mg ticagrelor or 10 mg prasugrel) was recommended for all patients for at least 12 months after DES implantation. Follow-up coronary angiography (CAG) and OCT were performed for evidence of myocardial ischemia in the stress test or clinical presentation of coronary artery disease (n = 84, 22.3%) or routine follow-up angiography (n = 293, 77.7%). Drug-coated balloon (DCB) angioplasty was performed for 80 patients who presented with ISR lesions, defined as lesions with diameter stenosis ≥ 70% on quantitative coronary analysis (QCA) or diameter stenosis ≥ 50% with the evidence of myocardial ischemia at the time of OCT. DCB angioplasty was not performed for lesions not consistent to ISR. When DCB angioplasty was required for the ISR lesions, OCT was performed before and after DCB angioplasty. The study protocol conformed to the ethical guidelines of the 1975 Declaration of Helsinki. The Institutional Review Board of Severance

Hospital, the Yonsei University approved the study protocols, and all participants provided written informed consent.

## Analysis of coronary angiography and OCT images

QCA was performed using an offline computerized QCA system (CAAS System; Pie Medical Imaging, Maastricht, The Netherlands) in an independent core laboratory (Cardiovascular Research Center, Seoul, Korea). OCT was performed using the C7-XR imaging system (Light-Lab Imaging; St. Jude Medical, St. Paul, MN, USA). All OCT images were analyzed using a certified offline software (QIvus; Medis Medical Imaging System, Leiden, The Netherlands) at a core laboratory (Cardiovascular Research Center) by analysts who were blinded to both the clinical and angiographic information. The cross-sectional OCT images were measured at 1-mm intervals for quantitative assessments. The stent and luminal cross-sectional area (CSA) were analyzed, and the neointimal CSA was calculated as the stent CSA minus the luminal CSA. The segment with minimal lumen area (MLA) and maximal neointimal proliferation was considered the representative site of lesions for future clinical follow-up [1, 2]. Therefore, the stented segments at the minimal lumen CSA and maximal neointimal CSA were qualitatively assessed to characterize the neointimal tissue as either homogeneous (a uniform signal-rich band without focal variation or attenuation), heterogeneous (focally changing optical properties and various backscattering patterns), or layered neointima (layers with different optical properties, namely, an adluminal high-scattering layer and an abluminal low-scattering layer) [4]. Neoatherosclerosis was defined as a lipid neointima (including a thin-cap fibroatheroma neointima, defined as a fibroatheroma with a fibrous cap <65 μm) or calcified neointima [5]. Although the OCT pattern had one of the three neointimal patterns (homogeneous, heterogeneous, or layered pattern) with at least one OCT feature of neoatherosclerosis, we classified it as neoatherosclerosis and excluded it from the current study. The neointimal morphologic characteristics were qualitatively evaluated by 2 observers who were blinded to the patients' clinical data and angiographic results. Inter- and intra-observer agreements for the assessment of the neointimal tissue characteristics in our core laboratory have been reported previously [6].

## DCB angioplasty and periprocedural OCT

All 80 patients who underwent DCB angioplasty at the time of OCT received 100 mg aspirin and 300 mg clopidogrel, 180 mg, ticagrelor or 60 mg prasugrel as the loading dose at 12 hours before DCB angioplasty. After diagnosis of ISR on angiography, pre-interventional OCT was performed before plain balloon dilation. Subsequently, paclitaxel-coated balloon (Sequent Please; B. Braun, Melsungen, Germany) was used for DCB angioplasty. The DCB size was determined based on the lesion length and stent diameter of ISR. The DCB was inflated for 60 seconds at the ISR lesion. Post-interventional OCT evaluation was conducted after DCB angioplasty. Dual anti-platelet (100 mg aspirin and 75 mg clopidogrel, 180 mg ticagrelor or 10 mg prasugrel) was prescribed for at least 1 month after DCB angioplasty.

## Follow-up

Events were assessed with a pre-specified protocol. All events were collected using a web-based reporting system. Additional information was obtained by medical records or telephone contact. The main endpoint was the incidence of non-target lesion revascularization (non-TLR) assessed according to the neointimal characteristics The TLR rate and any revascularization and myocardial infarction (MI) events were evaluated. TLR was defined as any repeat percutaneous intervention of the target lesion (including 5 mm proximal and 5 mm distal to the target

lesion) or surgical bypass of the target vessel performed for restenosis or other complications involving the target lesion. In this study, DES-treated lesions subjected to OCT were the target lesions. All lesions, except for DES-treated lesions subjected to OCT evaluation, were defined as non-target lesions [7]. MI was defined according to the third universal definition [8].

## Statistical analysis

Data are expressed as the number (%), mean ± standard deviation or median (interquartile range). Categorical data were compared using the chi-square test or Fisher's exact test. Continuous variables were compared using the Student's t-test for normally distributed data and Kruskal-Wallis H test for non-normally distributed data. Event-free survival was analyzed using the Kaplan-Meier survival curves and compared using the log-rank test between different groups. Using univariate Cox proportional hazards regression analysis, we analyzed 16 probable risk factors including age, sex, conventional cardiac risk factors, medication, stent generation, DCB angioplasty and neointimal characteristics. Age, sex and the variables with p-value <0.10 in the univariate analysis were included in the multivariate analysis to determine the independent predictors for revascularization. Univariate analysis using logistic regression was performed to identify independent predictors of the heterogeneous neointima formation. Age, sex and the variables with p-value less than 0.10 were entered in the multivariate analysis. A p-value <0.05 was considered statistically significant. Statistical analyses were performed using SPSS version 20.0.0 (IBM, Armonk, NY, USA).

## Results

### Baseline characteristics, angiography features, and QCA

The median duration between index PCI and OCT examination was 9.0 months (interquartile range: 6.0–13.5 months). CAG and OCT were performed for an evidence of ischemia or recurrent chest pain in 84 patients (acute coronary syndrome: 26 and stable angina: 58) and for routine follow up in 293 patients. DCB angioplasty was performed for 80 ISR lesions of the 377 stented lesions. The baseline, angiographic, and OCT characteristics of the patients according to the neointimal characteristics are shown in Table 1 [homogeneous (n = 207), heterogeneous (n = 93), and layered neointima (n = 77) (Table 1)].

Significant differences were not found over time from index PCI to OCT evaluation and over the clinical follow-up durations among the 3 neointima groups. The overall baseline and angiographic characteristics were comparable among the 3 neointimal groups. The patients in the heterogeneous and layered neointima groups were older than those in the homogeneous neointima group (p = 0.020). Although significant differences were not observed in the stent diameter and stent length, the heterogeneous and layered neointima groups had significantly greater diameter stenosis in QCA than the homogenous neointima group. In quantitative OCT analysis, the heterogeneous and layered neointima groups showed significantly smaller MLA than the homogenous neointima group. Based on the QCA and OCT results, DCB angioplasty was more frequently performed in the heterogeneous and layered neointima groups than in the homogenous neointima group (p<0.001).

In the QCA analysis at the time of OCT for non-TLR during the follow-up period, MLD and diameter stenosis were not different among the 3 neointimal groups. However, non-TLR developed significantly earlier in the heterogeneous neointima group than in the non-heterogeneous neointima group (mean duration from OCT to non-TLR: homogeneous vs. heterogeneous vs. layered 60.89 ± 30.56 vs. 32.92 ± 26.38 vs. 43.33 ± 33.96 months, p = 0.043; S1 Table).

**Table 1. Comparison of the baseline, angiographic, and OCT characteristics according to the neointimal characteristics.**

| Variables | Homogeneous neointima | Heterogeneous neointima | Layered neointima | p-value |
|---|---|---|---|---|
| | n = 207 (54.9%) | n = 93 (24.7%) | n = 77 (20.4%) | |
| Time from index PCI to OCT, months | 15.2 ± 18.1 | 18.4± 24.5 | 19.7 ± 22.0 | 0.191 |
| Duration of clinical follow-up, months | 39.5 ± 22.4 | 37.9 ± 22.1 | 41.5 ± 20.9 | 0.566 |
| Clinical characteristics | | | | |
| Age, years | 61.2 ± 9.7 | 63.6 ± 8.7 | 64.3 ± 8.6 | 0.020 |
| Male sex | 138 (66.7) | 63 (67.7) | 60 (77.9) | 0.177 |
| Diabetes | 63(30.4) | 34 (36.6) | 34 (44.2) | 0.089 |
| Hypertension | 124 (59.9) | 55 (59.1) | 55 (71.4) | 0.164 |
| Dyslipidemia | 99 (47.8) | 36 (38.7) | 39 (50.6) | 0.231 |
| Current smoker | 41 (19.8) | 23 (24.7) | 24 (31.2) | 0.124 |
| History of MI | 3 (1.4) | 4 (4.3) | 4 (5.2) | 0.164 |
| CKD | 2 (1.0) | 3 (3.2) | 0 (0) | 0.149 |
| Beta blocker | 171 (82.6) | 67 (72.0) | 60 (77.9) | 0.111 |
| ACEi or ARB | 145 (70.0) | 64 (68.8) | 54 (70.1) | 0.974 |
| Statin | 190 (91.8) | 84 (90.3) | 68 (88.3) | 0.661 |
| Angiographic profiles | | | | |
| Target vessel | | | | 0.079 |
| LAD | 116 (56.0) | 56 (60.2) | 44 (57.1) | |
| LCX | 49 (23.7) | 14 (15.1) | 9 (11.7) | |
| RCA | 42 (20.3) | 23 (24.7) | 24 (31.2) | |
| Stent types | | | | 0.506 |
| First-generation DES | 75 (36.2) | 29 (31.2) | 23 (29.9) | |
| Second-generation DES | 132 (63.8) | 64 (68.8) | 54 (70.1) | |
| Stent diameter, mm | 2.98± 0.33 | 3.05 ± 0.36 | 3.07 ± 0.40 | 0.088 |
| Stent length, mm | 23.9 ± 6.3 | 22.2 ± 6.9 | 23.7 ± 6.1 | 0.099 |
| Preinterventional QCA | | | | |
| Diameter stenosis, % | 27.0 ± 23.2 | 46.6 ± 31.7 | 50.6 ± 28.6 | <0.001 |
| Lesion length, mm | 17.1 ± 6.0 | 16.5 ± 7.0 | 15.0 ± 7.6 | 0.529 |
| DCB angioplasty | 22 (10.6) | 30 (32.3) | 28 (36.4) | <0.001 |
| OCT characteristics | | | | |
| Quantitative OCT profiles | | | | |
| (pre interventional) | | | | |
| MLA, mm$^2$ | 4.63 ± 1.66 | 4.13 ± 2.06 | 3.93 ± 2.11 | 0.007 |
| MSA, mm$^2$ | 5.11 ± 1.71 | 4.83 ± 1.47 | 4.88 ± 1.59 | 0.806 |
| Mean NIH area, mm$^2$ | 1.41 ± 1.03 | 1.20 ± 0.77 | 1.41 ± 0.92 | 0.224 |

Data are presented as mean ± standard deviation or number (%). ACEi, angiotensin-converting enzyme inhibitor; ARB, angiotensin receptor blocker; CKD, chronic kidney disease; DCB, drug-coated balloon; DES, drug-eluting stent; LAD, left anterior descending artery; LCX, left circumflex artery; MI, myocardial infarction; MLA, minimal lumenarea; MSA, minimal stent area; NIH, neointimal hyperplasia; OCT, optical coherence tomography; PCI, percutaneous coronary intervention; QCA, quantitative coronary angiography analysis; RCA, right coronary artery.

## Clinical outcomes

The median clinical follow-up duration after OCT examination was 40.0 months (interquartile range: 20.0–54.5 months). The incidence of TLR and any revascularization differed significantly according to the neointimal characteristics (p<0.001). The TLR rate was higher in order of the heterogeneous, layered, and homogeneous neointima groups; the differences in the TLR rates of each group were significant (homogeneous vs. heterogeneous vs. layered neointima:

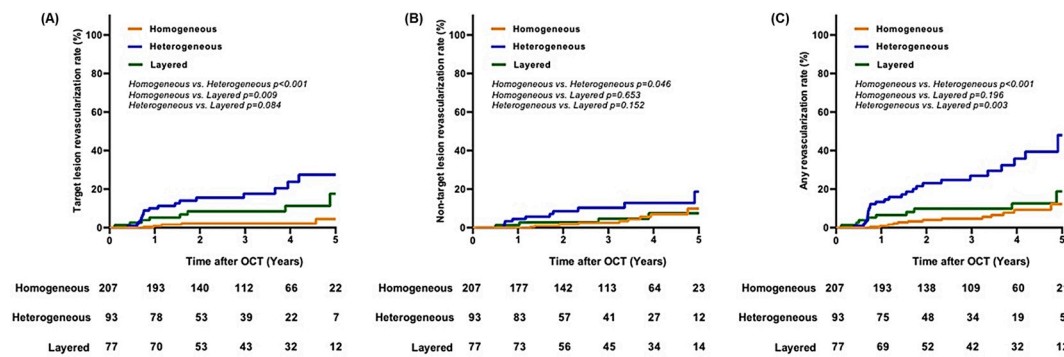

**Fig 1. Kaplan-Meier curves for the target lesion revascularization rate (A), non-target lesion revascularization rate (B), and any revascularization rate (C) according to the neointimal characteristics.**

2.9% vs. 19.4% vs. 10.4%, p<0.001; Fig 1A). The heterogeneous neointima group showed significantly higher non-TLR rate than the homogeneous neointima group and tended to have higher non-TLR rates than the layered neointima group (homogeneous vs. heterogeneous intima: 8.7% vs. 14.0%, p = 0.046; heterogeneous vs. layered neointima: 14.0% vs. 7.8%, p = 0.152; Table 2, Fig 1B). The heterogeneous neointima group had significantly higher any revascularization rate than the homogeneous and layered neointima groups (homogeneous vs. heterogeneous intima: 10.6% vs. 30.1%, p<0.001; heterogeneous vs. layered neointima: 30.1% vs. 13.0%, p = 0.003; Fig 1C).

TLR (36.7%) and non-TLR (23.3%) were more frequently observed in the patients who underwent DCB angioplasty in the heterogeneous neointima group (Fig 2A and 2B).

Subgroup analysis according to the reason for follow-up angiography was shown in S2 Table. The incidence of DCB angioplasty, heterogeneous neointima, any MI and TLR were significantly higher in group with an evidence of ischemia for follow-up angiography. However, there was no significant difference in the incidence of NTLR between group with an evidence of ischemia for follow-up angiography and routine follow-up angiography.

Multivariate analysis showed that heterogeneous neointima (HR: 2.237, 95% confidence interval [CI]: 1.023–4.890, p = 0.044) and chronic kidney disease (hazard ratio [HR]: 8.730, 95% CI: 2.175–35.036, p = 0.002) independently increased the non-TLR incidence (Table 3).

Heterogeneous neointima (HR: 2.671, 95% CI: 1.261–5.656, p = 0.010) and chronic kidney disease (HR: 5.971, 95% CI: 1.902–18.751, p = 0.002) were the independent predictors of any revascularization (Table 4). In addition, the significant predictor for the heterogeneous neointima was MI at the time of OCT (HR: 6.698, 95% CI: 1.212–37.022, p = 0.029) (S3 Table).

**Table 2. Clinical adverse events according to the neointimal characteristics at 40.0 months after OCT evaluation.**

|  | Homogeneous neointima | Heterogeneous neointima | Layered neointima | p-value |
|---|---|---|---|---|
|  | n = 207 (54.9%) | n = 93 (24.7%) | n = 77 (20.4%) |  |
| Non-TLR | 18 (8.7%) | 13 (14.0%) | 6 (7.8%) | 0.114 |
| Cardiac death | 4 (1.9%) | 1 (1.1%) | 0 (0%) | 0.588 |
| Any MI | 1 (0.5%) | 3 (3.2%) | 2 (2.6%) | 0.093 |
| TLR | 6 (2.9%) | 18 (19.4%) | 8 (10.4%) | <0.001 |
| Any revascularization | 22 (10.6%) | 28 (30.1%) | 10 (13.0%) | <0.001 |

Data are presented as number (%). MI, myocardial infarction; Non-TLR, non-target lesion revascularization; OCT, optical coherence tomography; TLR, target lesion revascularization

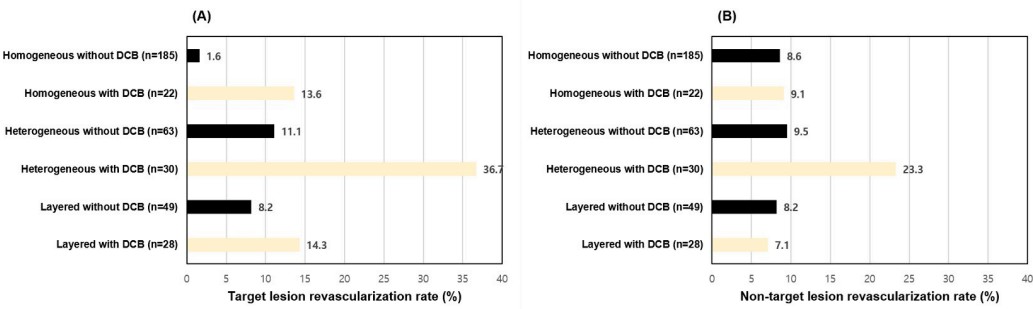

**Fig 2. Target lesion revascularization rates (A) and non-target lesion revascularization rates (B) according to the neointimal characteristics and DCB angioplasty.** DCB, drug-coated balloon.

## Discussion

In previous studies, heterogeneous neointima had been closely associated with higher target lesion failure rates in non-significant lesions not subjected to any interventions [1, 9] and significant ISR lesions treated with DCB compared to non-heterogeneous neointima [2, 3]. This study reported that 1) the incidence of TLR and non-TLR differed according to the in-stent neointimal characteristics and 2) patients with heterogeneous neointima with DESs were associated with a higher incidence of non-TLR and TLR.

Several studies have reported the prediction of cardiovascular events based on the OCT findings of de novo coronary artery. Thin-cap fibro-atheroma, lipid-rich plaque, and plaque rupture on OCT are the major risk factors and healed coronary plaque is a preventive factor for future coronary artery events [10–14]. However, only a few reports have described the

**Table 3. Independent predictors for non-target lesion revascularization.**

| Variable | Univariate analysis | | | Multivariate analysis | | |
|---|---|---|---|---|---|---|
| | HR | 95% CI | p-value | HR | 95% CI | p-value |
| Age | 0.988 | 0.956–1.022 | 0.487 | 0.985 | 0.947–1.023 | 0.432 |
| Male | 0.880 | 0.450–1.722 | 0.710 | 1.240 | 0.550–2.798 | 0.604 |
| Diabetes | 1.830 | 0.947–3.536 | 0.072 | 1.618 | 0.796–3.286 | 0.184 |
| Hypertension | 1.284 | 0.630–2.617 | 0.492 | | | |
| Dyslipidemia | 0.765 | 0.392–1.494 | 0.433 | | | |
| Current smoker | 1.496 | 0.735–3.048 | 0.267 | | | |
| CKD | 8.451 | 2.497–28.606 | 0.001 | 8.730 | 2.175–35.036 | 0.002 |
| History of MI | 3.210 | 0.756–13.632 | 0.114 | | | |
| DCB angioplasty after OCT | 1.317 | 0.615–2.823 | 0.478 | | | |
| First-generation DES | 0.986 | 0.489–1.991 | 0.970 | | | |
| Homogeneous neointima | 0.604 | 0.310–1.177 | 0.139 | | | |
| Heterogeneous neointima | 2.024 | 1.025–4.000 | 0.042 | 2.237 | 1.023–4.890 | 0.044 |
| Layered neointima | 0.852 | 0.349–2.082 | 0.726 | | | |
| ACEi or ARB | 0.891 | 0.416–1.907 | 0.766 | | | |
| Beta blocker | 0.538 | 0.269–1.078 | 0.080 | 0.901 | 0.387–2.096 | 0.809 |
| Statin | 1.004 | 0.303–3.325 | 0.995 | | | |

ACEi, angiotensin-converting enzyme inhibitor; ARB, angiotensin receptor antagonist; CKD, chronic kidney disease, CI, confidence interval; DCB, drug-coated balloon; DES, drug-eluting stent; MI, myocardial infarction; OCT, optical coherence tomography; HR, hazard ratio

**Table 4. Independent predictors for any revascularization.**

| Variable | Univariate analysis | | | Multivariate analysis | | |
|---|---|---|---|---|---|---|
| | HR | 95% CI | p-value | HR | 95% CI | p-value |
| Age | 0.994 | 0.968–1.020 | 0.633 | 0.978 | 0.950–1.007 | 0.132 |
| Male | 0.643 | 0.383–1.078 | 0.094 | 0.770 | 0.426–1.392 | 0.386 |
| Diabetes | 1.847 | 1.111–3.070 | 0.018 | 1.524 | 0.897–2.592 | 0.119 |
| Hypertension | 0.833 | 0.492–1.411 | 0.497 | | | |
| Dyslipidemia | 0.838 | 0.499–1.408 | 0.505 | | | |
| Current smoker | 1.259 | 0.716–2.213 | 0.424 | | | |
| CKD | 6.354 | 2.253–17.920 | <0.001 | 5.971 | 1.902–18.751 | 0.002 |
| History of MI | 1.439 | 0.349–5.927 | 0.614 | | | |
| DCB angioplasty after OCT | 2.430 | 1.406–4.203 | 0.001 | | | |
| First-generation DES | 1.043 | 0.609–1.788 | 0.877 | | | |
| Homogeneous neointima | 0.301 | 0.169–0.537 | <0.001 | 0.505 | 0.222–1.150 | 0.104 |
| Heterogeneous neointima | 4.324 | 2.537–7.370 | <0.001 | 2.671 | 1.261–5.656 | 0.010 |
| Layered neointima | 0.820 | 0.411–1.637 | 0.574 | | | |
| ACEi or ARB | 0.741 | 0.417–1.316 | 0.306 | | | |
| Beta blocker | 0.434 | 0.251–0.752 | 0.003 | 0.745 | 0.403–1.378 | 0.348 |
| Statin | 0.856 | 0.365–2.006 | 0.720 | | | |

ACEi, angiotensin-converting enzyme inhibitor; ARB, angiotensin receptor antagonist; CI, confidence interval; CKD, chronic kidney disease; DCB, drug-coated balloon; DES, drug-eluting stent; MI, myocardial infarction; OCT, optical coherence tomography; HR, hazard ratio

relationship between in-stent neointimal characteristics and non-target lesion outcome. Taniwaki et al. [15] reported the possible association between in-stent neoatherosclerosis and native coronary artery disease progression, suggesting similarities in the pathophysiologic mechanisms of the two entities. Our results also showed that the incidence of non-TLR was higher in the in-stent heterogeneous neointima group than in non-heterogeneous neointima groups, particularly homogeneous neointima group. Compared with a previous study, the current study focused on neointimal patterns without neoatherosclerotic features. This novel study suggests that the heterogeneous neointimal pattern without definite neoatherosclerosis may have clinical implications similar to those of neointima with neoatherosclerosis reported previously. Moreover, the change in neointimal pattern generally precedes neoatherosclerosis development, indicating that non-TLR events may be predicted earlier than that reported in a previous study with neoatherosclerosis. In terms of the possible mechanisms underlying the development of heterogeneous neointima, inflammation due to hypersensitivity reaction to drugs or polymers of a DES may play an important role in heterogeneous neointima growth and in native lesion progression [3]. In this study, the incidence of non-TLR in the heterogeneous neointima group with DCB angioplasty was higher than that of non-significant stented lesion with heterogeneous neointima as well as non-heterogeneous neointima with DCB angioplasty. This result indicates that atherosclerosis progression in non-stented de novo lesion may occur more rapidly in significant stented lesion with heterogeneous pattern than in non-significant lesion with heterogeneous neointima or significant lesion with homogeneous neointima. Given the important contribution of inflammatory processes to neointimal degeneration and native atherosclerosis as a continuity of coronary artery tree, the presence and extent of neointimal tissue inflammation may affect the atherosclerotic progression of de novo coronary artery that is not associated with stent [16, 17].

In previous articles, second-generation DES showed better results in intermediate-term strut apposition and coverage than first-generation DES, as well as superiority in long-term

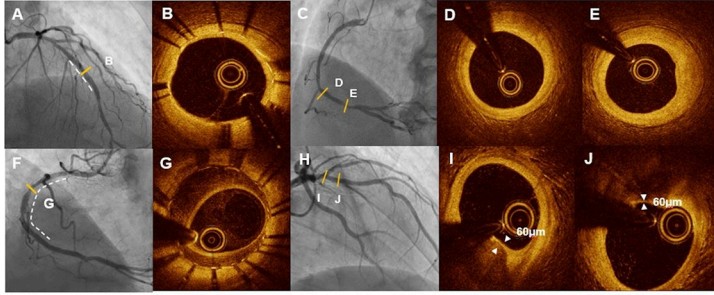

**Fig 3. Association of in-stent neointimal patterns and native coronary plaque characteristics.** Angiographic and optical coherence tomography images of the in-stent homogeneous neointimal pattern (A, B) and fibrous atherosclerotic plaque (C, D, E) and the in-stent heterogeneous neointimal pattern (F, G) and thin-cap fibroatheroma (H, I, J) in the native coronary artery.

clinical results [18, 19]. However, there have been little investigation dealt with the difference in DCB angioplasty outcomes between first-generation DES and second-generation DES. In present study, we did not find a significant difference of TLR rate between first-generation DES and second-generation DES (TLR rate; 8.7% vs 8.4%, p = 0.924). This finding may reflect similar efficacy of DCB angioplasty either first-generation or second-generation DES ISR, but this result should be evaluated with further clinical trials.

The risk factor of heterogeneous neointima formation is not well known. In present study, clinical presentation of MI at the time of OCT was independent risk factors for the development of heterogeneous neointimal tissue. Gonzalo et al. [4] also demonstrated that the incidence of heterogeneous neointima is much higher in patients with unstable angina than those with stable angina. In our study, in patients with non-TLR, a similar degree of stenosis was noted among the 3 neointimal patterns at the time of OCT. However, non-TLR occurrence developed earlier in the in-stent heterogeneous group, which may suggest a rapid progression of native lesion in this group of neointimal pattern. A possible explanation was shown in recent cases with OCT evaluation of both stented and native lesions (these cases were not included in the current analysis; Fig 3).

Although whole vessel OCT examination is expected to provide more accurate information to assess the association between the stented and native lesions and predict future coronary events, performing OCT on all coronary arteries is impractical because it requires an excessive use of contrast agents and increases procedure time. Otherwise, an OCT for stented lesions may be a more clinically useful method to provide information on lesion progression in native lesion as well as stented lesion, which can be more practical approach in real clinical practice.

To date, little or no treatment has been known to affect the formation and changes in in-stent neointimal characteristics. However, a previous study [20] reported that an intensive lipid control may be beneficial to prevent heterogeneous neointimal degeneration. In the current clinical viewpoint, applicable treatment options are limited to reduce a future event in patients with in-stent heterogeneous neointima, and potent lipid-lowering agents or anti-inflammatory agents is a possible suggested approach. However, more investigation and data are warranted to support this concept.

This study has several limitations. First, although the neointima was categorized based on frames within MLA or greatest neointimal CSA, a single category may not sufficiently represent the neointima when the lesion was diffused. Second, the study had a heterogeneous population because we included all patients who underwent an OCT examination regardless of

DCB treatment. Third, time from index PCI to OCT was not constant, and variation among individuals was noted. Nevertheless, follow-up duration was adequate to assess the effect of neointimal characteristics and not different among the 3 neointima groups. Fourth, a neointimal pattern with neoatherosclerosis was not included in this analysis which may require further study. Finally, even if we provided representative OCT images of the association between in-stent neointimal pattern and native coronary artery plaque pattern, we did not perform OCT on all of the non-target lesions. Therefore, we could not completely reveal the difference of the OCT pattern of non-target lesion depending on the in-stent neointimal characteristics.

## Conclusions

Heterogeneous neointima in DES-treated lesion was associated with a higher incidence of non-TLR and stented target lesion failure. This finding suggests that the neointimal pattern of DES may be a possible prognostic indicator of native lesion progression.

## Supporting information

**S1 Table. Quantitative coronary analysis of non-stented lesions occurred a non-target lesion revascularization at time of OCT evaluation.**
(DOCX)

**S2 Table. Subgroup analysis according to the reason for follow-up angiography.**
(DOCX)

**S3 Table. Independent predictors for heterogeneous neointima.**
(DOCX)

## Author Contributions

**Conceptualization:** Hae Won Jung, Chewan Lim, Han Joon Bae, Jung-Sun Kim, Seung-Jun Lee, Byeong-Keuk Kim, Donghoon Choi, Myeong-Ki Hong, Yangsoo Jang.

**Data curation:** Hae Won Jung, Chewan Lim, Jung-Sun Kim, Seung-Jun Lee.

**Formal analysis:** Han Joon Bae, Jung-Sun Kim, Seung-Jun Lee, Chul-Min Ahn.

**Funding acquisition:** Jung-Sun Kim.

**Investigation:** Hae Won Jung, Chewan Lim, Jung-Hee Lee, Yong-Joon Lee, Jung-Sun Kim, Chul-Min Ahn.

**Methodology:** Hae Won Jung, Chewan Lim, Jung-Hee Lee, Yong-Joon Lee, Young-Guk Ko.

**Project administration:** Chul-Min Ahn.

**Software:** Sung-Jin Hong.

**Validation:** Jung-Hee Lee.

**Visualization:** Sung-Jin Hong.

**Writing – original draft:** Hae Won Jung, Chewan Lim, Jung-Sun Kim.

**Writing – review & editing:** Han Joon Bae, Yong-Joon Lee, Jung-Sun Kim, Sung-Jin Hong, Byeong-Keuk Kim, Young-Guk Ko, Donghoon Choi, Myeong-Ki Hong, Yangsoo Jang.

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
