## [Decision Letter · Decision Letter 0]

25 Nov 2020

PONE-D-20-31810

Association between in-stent neointimal characteristics and native coronary artery disease progression

PLOS ONE

Dear Dr. Kim,

Thank you for submitting your manuscript to PLOS ONE. After careful consideration, we feel that it has merit but does not fully meet PLOS ONE’s publication criteria as it currently stands. Therefore, we invite you to submit a revised version of the manuscript that addresses the points raised during the review process.

We look forward to receiving your revised manuscript.

Kind regards,

Salvatore De Rosa

Academic Editor

PLOS ONE

Additional Editor Comments (if provided):

The authors present interesting data on the use of OCT as a marker of clinical risk in patients with in-stent restenosis. They included patients that had received DESs of different generations. It is known that in-stent restenosis and neointima characteristics may present specific features in first- versus newer-generations of DES (Circ Cardiovasc Interv. 2015;8(4):e002375. doi: 10.1161/CIRCINTERVENTIONS.115.002375.). It is interesting that despite those differences, OCT-assessd heterogeneous neointima seems to be a prognostic factos independently from the DES generation. Please comment on this specific results in the discussion.

Reviewers' comments:

Reviewer's Responses to Questions

**Comments to the Author**

1. Is the manuscript technically sound, and do the data support the conclusions?

Reviewer #1: Yes

Reviewer #2: Yes

2. Has the statistical analysis been performed appropriately and rigorously? 

Reviewer #1: Yes

Reviewer #2: No

3. Have the authors made all data underlying the findings in their manuscript fully available?

Reviewer #1: Yes

Reviewer #2: Yes

4. Is the manuscript presented in an intelligible fashion and written in standard English?

Reviewer #1: Yes

Reviewer #2: Yes

5. Review Comments to the Author

Reviewer #1: The present is a very interesting study and the authors should be complimented for.

Methods:

- subgroup analysis for patients performing re-angio for ischemia or for angiographic follow up should be performed

- do authors have a core-lab?

- performing OCT in tight lesions is not always so feasible. How many times did authors not manage to perform OCT?

- authors should perform a multivariate analysis to evaluate the predictors of different kind of neo-atherosclerosis

Results and discussions: different rates of ISR have been described according to stent type (quote on PMID: 27099274). This should be evalauted or at least quoted in discussion.

Reviewer #2: The current manuscript summarizes the findings of a moderately-sized observational retrospective OCT imaging study, which aims to investigate the relationship between in-stent neointimal characteristics and progression of native atherosclerosis. The neointimal characteristics of 377 patients with 377 drug-eluting stents (DESs) were quantitatively and qualitatively assessed using OCT. The OCT-based neointima was categorized as homogeneous (n=207), heterogeneous (n=93), and layered (n=77). The heterogeneous neointima in DES-treated lesions was associated with a higher incidence of non-TLR and target lesion failure.

The study is of interest since it relates to a topic of interest to the readership of this Journal, it is well written, mostly with appropriate methodology. Some concerns remain however:

The authors

1. The authors mentioned that Drug-coated balloon (DCB) angioplasty was performed for 80 patients who presented with in-stent restenosis (ISR) lesions, defined as lesions with diameter stenosis ≥ 70% on quantitative coronary analysis (QCA) or diameter stenosis ≥ 50% with the evidence of myocardial ischemia at the time of OCT.

Have you not performed the DES in DES strategy in any lesions? I think that in some lesions we need the stent implantation. Why you have not do that strategy in any lesions even though several guidelines recommend both DES and DCB strategy to the ISR lesions.

2. The most major problem was that your inclusion criteria was really complex. I had a big concern that the authors included both DES lesions to be treated (ISR) and not to be treated (no ISR). Obviously you had a huge selection bias. I think you should not investigate the TLR rate and even no-TLR rate for the lesion with ISR and no ISR together. Please figure out.

3. The authors excluded the lesions with neoatherosclerosis. I want to know the theoretical reason to exclude that. Generally, after the implantation of especially DES, neoatherosclerosis was the important problem even in the newgeneration DES. Some papers demonstrated that the DES with neoatherosclerosis had worse clinical outcome compared with no noatherosclerosis.

4. The definition of neoatherosclerosis was a little bit tricky. I am confused the difference of ≥ 2 quadrants with lipid content and the thinnest part of the fibrous cap measuring ≤ 65μm, and thin-cap fibroatheroma.

5. The patient selection part was relatively difficult to be understood. Please clarify this part and do not repeat the DES treated lesion.

6. For the multivariate analysis of non-TLR, what factors you included? Please clarify that in the method section.

7. How were events recorded and assessed over time? Was there prespecified protocol or timing?

6. PLOS authors have the option to publish the peer review history of their article (what does this mean?). If published, this will include your full peer review and any attached files.

Reviewer #1: **Yes: **Fabrizio D'Ascenzo

Reviewer #2: **Yes: **Daisuke Nakamura

---

## [Author Response · Author response to Decision Letter 0]

4 Jan 2021

Response Letter to Editor-in-Chief 

We deeply appreciate your time and input in reviewing our manuscript. We have carefully considered the comments given by the editorial board, and have addressed them point-by-point as below.

Reviewer #1

1. Subgroup analysis for patients performing re-angio for ischemia or for angiographic follow up should be performed.

Author’s response

We agree with the Reviewer’s opinion. We performed a follow-up angiography for the evidence of ischemia or recurrent chest pain in 84 patients and a routine angiographic follow-up in 293 patients. Subgroup analysis according to the reason for follow-up angiography was added in Table S2. The incidence of DCB angioplasty, heterogeneous neointima and TLR were significantly higher in group with an evidence of ischemia for follow-up angiography. However, there was no significant difference in the incidence of NTLR between group with an evidence of ischemia for angiography and routine follow-up angiography.

Method, Patient data selection, page 4

Follow-up coronary angiography (CAG) and OCT were performed for evidence of myocardial ischemia in the stress test or clinical presentation of coronary artery disease (n=84, 22.3%) or routine follow-up angiography (n=293, 77.7%).

Results, Baseline characteristics, angiography features, and QCA, page 7

The median clinical follow-up duration was 9.0 months (interquartile range: 6.0–13.5 months). CAG and OCT were performed for an evidence of ischemia or recurrent chest pain in 84 patients (acute coronary syndrome: 26 and stable angina: 58) and for routine follow up in 293 patients. DCB angioplasty was performed for 80 ISR lesions of the 377 stented lesions. The baseline, angiographic, and OCT characteristics of the patients according to the neointimal characteristics are shown in Table 1 [homogeneous (n=207), heterogeneous (n=93), and layered neointima (n=77) (Table 1)].

Results, Clinical outcomes, page 13

Subgroup analysis according to the reason for follow-up angiography was added in Table S2. The incidence of DCB angioplasty, heterogeneous neointima, any MI and TLR were significantly higher in group with an evidence of ischemia for follow-up angiography. However, there was no significant difference in the incidence of NTLR between group with an evidence of ischemia for follow-up angiography and routine follow-up angiography.

2. Do authors have a core-lab?

Author’s response

We have an independent core laboratory (Cardiovascular Research Center, Seoul, Korea) and database in Yonsei OCT registry (ClinicalTrials.gov, NCT02099162). We have analyzed OCT images through our core laboratory with imaging experts. 

Material and Methods, Page 4

All OCT images were analyzed using a certified offline software (QIvus; Medis Medical Imaging System, Leiden, The Netherlands) at a core laboratory (Cardiovascular Research Center) by analysts who were blinded to both the clinical and angiographic information.

3. Performing OCT in tight lesions is not always so feasible. How many times did authors not manage to perform OCT?

Author’s response

In our study, when DCB angioplasty was required for the ISR lesions, OCT was performed before and after DCB angioplasty. However, there were 5 cases among 576 cases which was not performed OCT examination before balloon angioplasty because OCT catheter did not pass due to tight stenosis. In these 5 cases, OCT was performed after balloon angioplasty. These 5 cases were excluded an analysis from this study. We have added the above to the exclusion criteria of this study.

Method, Patient data selection, page 3

Among 540 patients with 576 stented lesions from the Yonsei OCT registry (Clinical Trials.gov NCT02099162), we excluded 163 patients from the study for the following reasons: 1) inadequate OCT image quality (n=5), 2) index PCI with bare metal stents (n=45), 3) OCT for two or more stented lesions (n=36), 4) OCT after balloon angioplasty due to tight lesion (n=5), 5) OCT-based evidence of neoatherosclerotic lesions (n=39), 6) in-stent restenosis (ISR) lesions treated with a DES (n=30) and 7) loss to follow-up (n=3).

4. The authors should perform a multivariate analysis to evaluate the predictors of different kind of neo-atherosclerosis

Author’s response

We appreciate your comprehensive comment. In our study, heterogeneous neoinitma was the major predictor of TLR and non-TLR. Therefore, we added the result to evaluate the predictors of heterogeneous neointima in Table S3. We also performed the analysis for predictors for homogeneous and layered neointima as your suggestion. Younger age were independent predictors for homogeneous neointima. Old age and male were independent predictors for layered neointima. But we did not include the manuscript.

Method, statistical analysis, page 7 

Univariate analysis using logistic regression was performed to identify independent predictors of the heterogeneous neointima formation. Age, sex and variables achieving a p-value less than 0.10 were entered in the multivariate analysis.

Results, page 14

In addition, the significant predictor for the heterogeneous neointima was MI at the time of OCT (HR: 6.698, 95% CI: 1.212–37.022, p=0.029) (Table S3).

5. Results and discussions: different rates of ISR have been described according to stent type (quote on PMID: 27099274). This should be evaluated or at least quoted in discussion.

Author’s response

We agree with the Reviewer’s opinion. Second-generation DES showed better results in intermediate-term strut apposition and coverage than first-generation DES, as well as superiority in long-term clinical results. However, few papers have dealt with the difference in DCB angioplasty outcomes between first-generation DES and second-generation DES. 

In present study, we did not find significant differences of TLR rate between first-generation generation DES and second-generation generation DES. (TLR rate; first-generation DES vs second-generation: 8.7% vs 8.4%, p=0.924). We added this issue to the discussion.

Discussion, page 17

In previous articles, second-generation DES showed better results in intermediate-term strut apposition and coverage than first-generation DES, as well as superiority in long-term clinical results [18,19]. However, there have been little investigation dealt with the difference in DCB angioplasty outcomes between first-generation DES and second-generation DES. In present study, we did not find a significant difference of TLR rate between first-generation DES and second-generation DES (TLR rate; 8.7% vs 8.4%, p=0.924). This finding may reflect similar efficacy of DCB angioplasty either first-generation or second-generation DES ISR, but this result should be evaluated with further clinical trials. 

References

18. Iannaccone M, D'Ascenzo F, Templin C, Omede P, Montefusco A, Guagliumi G, et al. Optical coherence tomography evaluation of intermediate-term healing of different stent types: systemic review and meta-analysis. Eur Heart J Cardiovasc Imaging 2017;18(2):159-66.

19. Palmerini T, Benedetto U, Biondi-Zoccai G, Della Riva D, Bacchi-Reggiani L, Smits PC, et al. Long-Term Safety of Drug-Eluting and Bare-Metal Stents: Evidence From a Comprehensive Network Meta-Analysis. J Am Coll Cardiol 2015;65(23):2496-507.

Reviewer #2

1. The authors mentioned that Drug-coated balloon (DCB) angioplasty was performed for 80 patients who presented with in-stent restenosis (ISR) lesions, defined as lesions with diameter stenosis ≥ 70% on quantitative coronary analysis (QCA) or diameter stenosis ≥ 50% with the evidence of myocardial ischemia at the time of OCT. Have you not performed the DES in DES strategy in any lesions? I think that in some lesions we need the stent implantation. Why you have not done that strategy in any lesions even though several guidelines recommend both DES and DCB strategy to the ISR lesions.

Author’s response

We agree with the Reviewer’s opinion. DES is one of the good treatment options for the ISR lesion. However, our center still prioritizes DCB as a treatment for ISR lesions, and DES is used when ISR recurs or when dissection occurs after DCB angioplasty. Therefore, DES treatment for ISR has been done in a relatively small number of patients in our center (n=30, recurrent ISR; n=5, edge dissection; n=18. insufficient expansion of ISR lesions after balloon angioplasty; n=7). In this analysis, we excluded 30 patients treated with DES for ISR lesions. 

Materials and methods, Patient data selection, Page 3

Among 540 patients with 576 stented lesions from the Yonsei OCT registry (Clinical Trials.gov NCT02099162), we excluded 163 patients from the study for the following reasons: 1) inadequate OCT image quality (n=5), 2) index PCI with bare metal stents (n=45), 3) OCT for two or more stented lesions (n=36), 4) OCT after balloon angioplasty due to tight lesion (n=5), 5) OCT-based evidence of neoatherosclerotic lesions (n=39), 6) in-stent restenosis (ISR) lesions treated with a DES (n=30) and 7) loss to follow-up (n=3).

2. The most major problem was that your inclusion criteria were really complex. I had a big concern that the authors included both DES lesions to be treated (ISR) and not to be treated (no ISR). Obviously, you had a huge selection bias. I think you should not investigate the TLR rate and even non-TLR rate for the lesion with ISR and no ISR together. Please figure out.

Author’s response

We agree with the Reviewer’s opinion that current study included both DES lesions to be treated (ISR) and not to be treated (no ISR), and these 2 lesions may have different characteristics. Due to limited number of populations, we tried to include all population both ISR and no ISR and evaluated the incidence of non-target lesion revascularization as a main outcome. Therefore, it may be reasonable to include all population treated by drug eluting stent before, but we also included this issue in limitation.

Discussion, Limitation, page 19

This study has several limitations. First, although the neointima was categorized based on frames within MLA or greatest neointimal CSA, a single category may not sufficiently represent the neointima when the lesion was diffused. Second, the study had a heterogeneous population because we included all patients who underwent an OCT examination regardless of DCB treatment. 

3. The authors excluded the lesions with neoatherosclerosis. I want to know the theoretical reason to exclude that. Generally, after the implantation of especially DES, neoatherosclerosis was the important problem even in the new generation DES. Some papers demonstrated that the DES with neoatherosclerosis had worse clinical outcome compared with no neoatherosclerosis.

Author’s response

We deeply agree with your opinion. As you have pointed out, patients with neoatherosclerosis on OCT reported to have a poor prognosis compared to patients without neoatherosclerosis. Neoatherosclerosis can be detected often after a long period of time after stent implanation. According to the report of Kim C et al., the detection frequency of neoatherosclerosis was just 6.4% before 1 year of DES insertion. [Kim C, et al. Am Heart J. 2015. PMID: 26385044]. In our study, the period from stent insertion to OCT was considerably short as median 9.0 months (interquartile range: 6.0–13.5 months). For this reason, neoatherosclerosis was found in very few patients. We identified 39 lesions of neoatherosclerosis out of 576 lesions (6.77%). Although Taniwaki et al. suggested possible association between in-stent neoatherosclerosis and native coronary artery disease progression [Taniwaki M, et al. Eur Heart J. 2015. PMID: 26040806], study follow-up period of current study was relatively shorter to assess the neoatherosclerosis related with non-TLR. Therefore, we tried to focus on early association with non-TLR and early neointimal change such as homogeneous, heterogeneous or layered rather than late neointimal change such as neoatherosclerosis. The changes in neointimal pattern generally precede neoatherosclerosis development, indicating that cardiac events may be predicted earlier than that reported in a previous study with neoatherosclerosis. This issue was addressed on discussion.

Discussion, Page 16

Compared with a previous study, the current study focused on neointimal patterns without neoatherosclerotic features. This novel study suggests that the heterogeneous neointimal pattern without definite neoatherosclerosis may have clinical implications similar to those of neointima with neoatherosclerosis reported previously. Moreover, the change in neointimal pattern generally precedes neoatherosclerosis development, indicating that non-TLR events may be predicted earlier than that reported in a previous study with neoatherosclerosis.

Limitations, Page 19

Fourth, a neointimal pattern with neoatherosclerosis was not included in this analysis and needed its clinical implication with further study.

4. The definition of neoatherosclerosis was a little bit tricky. I am confused the difference of ≥ 2 quadrants with lipid content and the thinnest part of the fibrous cap measuring ≤ 65μm, and thin-cap fibroatheroma.

Author’s response

We apologize for your confusion due to the ambiguous expression. We have revised the description of the neoatherosclerosis characteristics and changed the reference.

Methods, Analysis of coronary angiography and OCT images, page 5

Neoatherosclerosis was defined as a lipid neointima (including a thin-cap fibroatheroma neointima, defined as a fibroatheroma with a fibrous cap <65 µm) or calcified neointima [5]. 

References

5. Nakamura D, Yasumura K, Nakamura H, Matsuhiro Y, Yasumoto K, Tanaka A, et al. Different Neoatherosclerosis Patterns in Drug-Eluting- and Bare-Metal Stent Restenosis- Optical Coherence Tomography Study. Circ J. 2019;83(2):313-9.

5. The patient selection part was relatively difficult to be understood. Please clarify this part and do not repeat the DES treated lesion.

Author’s response

Thanks for your kind comment. As your comment, we modified the patient selection part to make it clear for readers to understand.

Materials and methods, Patient data selection, Page 3

Among 540 patients with 576 stented lesions from the Yonsei OCT registry (Clinical Trials.gov NCT02099162), we excluded 163 patients from the study for the following reasons: 1) inadequate OCT image quality (n=5), 2) index PCI with bare metal stents (n=45), 3) OCT for two or more stented lesions (n=36), 4) OCT after balloon angioplasty due to tight lesion (n=5), 5) OCT-based evidence of neoatherosclerotic lesions (n=39), 6) in-stent restenosis (ISR) lesions treated with a DES (n=30) and 7) loss to follow-up (n=3).

6. For the multivariate analysis of non-TLR, what factors you included? Please clarify that in the method section.

Author’s response

 Thank you for your comment. We clarified the factors included in Cox proportional hazards regression analysis.

Methods, Statistical analysis, page 7

Using univariate Cox proportional hazards regression analysis, we analyzed 16 probable risk factors including age, sex, conventional cardiac risk factors, medication, stent generation, DCB angioplasty and neointimal characteristics. Age, sex and variables achieving a p-value <0.10 in the univariate analysis were included in the multivariate analysis to determine the independent predictors for revascularization.

7. How were events recorded and assessed over time? Was there prespecified protocol or timing?

Author’s response

Events record and assess were proceeded with a pre-specified protocol in Yonsei OCT registry (Clinical Trials.gov NCT02099162). All events were collected using a web-based reporting system. Additional information was obtained by medical records or telephone contact. We added the description in detail. 

Methods, Follow-up, Page 6 

Events were assessed with a pre-specified protocol. All events were collected using a web-based reporting system. Additional information was obtained by medical records or telephone contact.

---

## [Decision Letter · Decision Letter 1]

8 Feb 2021

Association between in-stent neointimal characteristics and native coronary artery disease progression

PONE-D-20-31810R1

Dear Dr. Kim,

We’re pleased to inform you that your manuscript has been judged scientifically suitable for publication and will be formally accepted for publication once it meets all outstanding technical requirements.

Kind regards,

Salvatore De Rosa

Academic Editor

PLOS ONE

Additional Editor Comments (optional):

Reviewers' comments:

Reviewer's Responses to Questions

**Comments to the Author**

1. If the authors have adequately addressed your comments raised in a previous round of review and you feel that this manuscript is now acceptable for publication, you may indicate that here to bypass the “Comments to the Author” section, enter your conflict of interest statement in the “Confidential to Editor” section, and submit your "Accept" recommendation.

Reviewer #1: All comments have been addressed

Reviewer #2: All comments have been addressed

2. Is the manuscript technically sound, and do the data support the conclusions?

Reviewer #1: (No Response)

Reviewer #2: Yes

3. Has the statistical analysis been performed appropriately and rigorously? 

Reviewer #1: (No Response)

Reviewer #2: Yes

4. Have the authors made all data underlying the findings in their manuscript fully available?

Reviewer #1: (No Response)

Reviewer #2: Yes

5. Is the manuscript presented in an intelligible fashion and written in standard English?

Reviewer #1: (No Response)

Reviewer #2: Yes

6. Review Comments to the Author

Reviewer #1: (No Response)

Reviewer #2: All my questions and comments had been accurately answered.

I agree that this paper should be accepted. Thank you for your all effort.

7. PLOS authors have the option to publish the peer review history of their article (what does this mean?). If published, this will include your full peer review and any attached files.

Reviewer #1: **Yes: **Fabrizio D'Ascenzo

Reviewer #2: No

---

## [Editor Report · Acceptance letter]

12 Apr 2021

PONE-D-20-31810R1 

Association between in-stent neointimal characteristics and native coronary artery disease progression 

Dear Dr. Kim:

I'm pleased to inform you that your manuscript has been deemed suitable for publication in PLOS ONE. Congratulations! Your manuscript is now with our production department. 

Kind regards, 

on behalf of

Dr. Salvatore De Rosa 

Academic Editor

PLOS ONE